# The GALANT trial: study protocol of a randomised placebo-controlled trial in patients with a $^{68}$Ga-DOTATATE PET-positive, clinically non-functioning pituitary macroadenoma on the effect of lanreotide on tumour size

Tessel M Boertien ![ORCID],[1] Madeleine L Drent,[2] Jan Booij,[3] Charles B L M Majoie,[3] Marcel P M Stokkel,[4] Jantien Hoogmoed,[5] Alberto Pereira,[6] Nienke R Biermasz,[6] Suat Simsek,[2,7] Ronald Groote Veldman,[8] Michael W T Tanck,[9] Eric Fliers,[1] Peter H Bisschop[1]

EF and PHB contributed equally.

For numbered affiliations see end of article.

**Correspondence to**
Tessel M Boertien;
t.m.boertien@amsterdamumc.nl

## ABSTRACT

**Introduction** At present, there is no approved medical treatment option for patients with non-functioning pituitary adenoma. A number of open-label studies suggest that treatment with somatostatin analogues may prevent tumour progression. In vivo somatostatin receptor imaging using $^{68}$Ga-DOTATATE PET (PET, positron emission tomography) could help in preselecting patients potentially responsive to treatment. Our aim is to investigate the effect of the somatostatin analogue lanreotide as compared with placebo on tumour size in patients with a $^{68}$Ga-DOTATATE PET-positive non-functioning pituitary macroadenoma (NFMA).

**Methods and analysis** The GALANT study is a multicentre, randomised, double-blind, placebo-controlled trial in adult patients with a suprasellar extending NFMA. Included patients undergo a $^{68}$Ga-DOTATATE PET/CT of the head and tracer uptake is assessed after coregistration with pituitary MRI. Forty-four patients with a $^{68}$Ga-DOTATATE PET-positive NFMA are randomised in a 1:1 ratio between lanreotide 120 mg or placebo, both administered as subcutaneous injections every 28 days for 72 weeks. The primary outcome is the change in cranio-caudal tumour diameter on pituitary MRI after treatment. Secondary outcomes are change in tumour volume, time to tumour progression, change in quality of life and number of adverse events. Final results are expected in the second half of 2021.

**Ethics and dissemination** The study protocol has been approved by the Medical Research Ethics Committee of the Academic Medical Centre (AMC) of the Amsterdam University Medical Centres and by the Dutch competent authority. It is an investigator-initiated study with financial support by Ipsen Farmaceutica BV. The AMC, as sponsor, remains owner of all data. Results will be submitted for publication in a peer-reviewed journal.

**Trial registration number** NL5136 (Netherlands Trial Register); pre-recruitment.

### Strengths and limitations of this study

► The GALANT study is the first double-blind and placebo-controlled intervention trial in non-functioning pituitary macroadenoma (NFMA) patients.
► $^{68}$Ga-DOTATATE PET/CT (PET, positron emission tomography) is used to select only those patients with a somatostatin receptor-positive adenoma for randomisation between lanreotide and placebo.
► The study is powered on change in cranio-caudal adenoma diameter, a clinically relevant outcome in the prevention of optic chiasm compression.
► The exclusion of patients previously treated with radiotherapy may present a bias in this study towards NFMA with a less aggressive growth rate.

## INTRODUCTION

Pituitary adenomas are benign tumours of the pituitary gland and are recognised as the third most common intracranial neoplasm.[1] Clinically non-functioning adenomas (NFA) account for 15%–50% of all pituitary adenomas.[2] Because of the lack of clinical and biochemical signs of hormonal hypersecretion, NFA can go undetected for a long time and usually are macroadenomas, that is, diameter ≥1 cm, at the time of diagnosis. Presenting symptoms are related to mass effects and include pituitary insufficiency, headache and visual disturbances.[2]

Transsphenoidal resection of the adenoma is the main therapeutic approach. However, the overall remission rate after surgery is only 44%, and regrowth of residual tumour occurs in over 50% of patients.[3,4] Consequently,

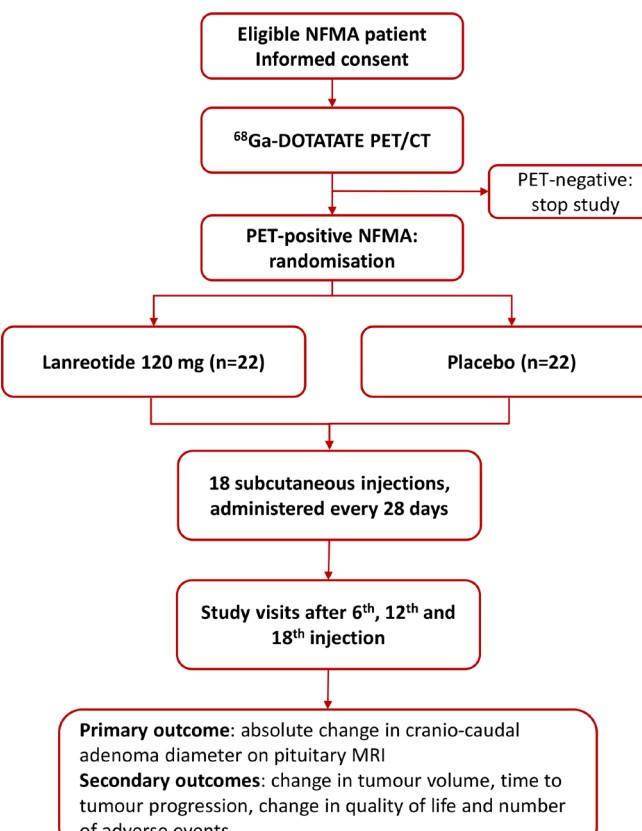

**Figure 1** Flowchart of the GALANT (a randomised placebo-controlled trial in patients with a [68]Ga-DOTATATE PET-positive, clinically NFMA on the effect of lanreotide on tumour size) study. NFMA, non-functioning pituitary macroadenoma; PET, positron emission tomography.

a substantial part of patients requires additional treatment during follow-up. Repeat surgery is recommended in case of persistent or recurrent chiasm compression, but the benefit/risk ratio is less favourable than with primary surgery.[5 6] Adjuvant radiotherapy is effective in controlling tumour (re)growth, but its use is restricted in view of the high risk of hypopituitarism and rare but important complications such as optic nerve damage and secondary brain tumours.[7–9]

Effective medical therapy as alternative option in NFA management would therefore be of great value. The expression of somatostatin receptor (SSTR) subtypes in the majority of NFA provides a potential target for receptor-mediated therapy.[10–16] The somatostatin analogues (SSA) octreotide and lanreotide have highest affinity for $SSTR_2$ and $SSTR_5$, and these subtypes are implicated in antiproliferative effects.[17] Indeed, treatment with octreotide seems to prevent tumour progression in selected NFA patients, although interpretation is hampered by methodological heterogeneity among studies and open-label design.[10 18]

SSA could thus play a role in the management of NFA patients, but outcomes of well-designed placebo-controlled studies are lacking. A promising approach may be in vivo SSTR imaging to preselect patients potentially

responsive to treatment.[10 19] Thus far, SSTR expression in NFA has been assessed through [111]In-DTPA-octreotide scintigraphy or single-photon emission computed tomography (SPECT).[10 14 15 20] However, interpretation of pituitary adenoma uptake using scintigraphy/SPECT is limited by low spatial resolution.[21] The introduction of the positron emission tomography (PET) radiotracer [68]Ga-DOTATATE has enabled high-resolution PET/CT imaging with additional superb $SSTR_2$ affinity.[22 23]

The aim of the present study is to investigate the effect of the SSA lanreotide, as compared with placebo, on adenoma size in patients with a [68]Ga-DOTATATE PET-positive non-functioning pituitary macroadenoma (NFMA).

## METHODS AND ANALYSIS

This manuscript was written following the Standard Protocol Items: Recommendations for Interventional Trials guideline on reporting of intervention trial protocols (online supplementary material S1).[24 25]

### Study design and setting

The GALANT study is a multicentre, randomised, double-blind, placebo-controlled, parallel-group, superiority trial in patients with a [68]Ga-DOTATATE PET-positive NFMA, investigating the effect of lanreotide versus placebo on tumour size. Forty-four patients are randomised in a 1:1 ratio and the primary outcome measure is the absolute change in cranio-caudal adenoma diameter after 72 weeks of treatment (study flowchart in figure 1). The study is conducted in an outpatient setting and eligible patients are referred by endocrinologists at academic and non-academic hospitals in the Netherlands for inclusion at one of the participating centres (Amsterdam University Medical Centres (locations AMC and VUMC) and Leiden University Medical Centre). The study-related [68]Ga-DOTATATE PET/CT is performed at either the AMC or the Netherlands Cancer Institute in Amsterdam, an imaging-only participating centre. The study is investigator-initiated with the AMC as sponsor. Financial support is provided by Ipsen Farmaceutica BV.

### Eligibility criteria

Adult patients diagnosed with an NFMA with suprasellar extension, either surgery-naïve or with a postoperative remnant, are eligible for inclusion. Detailed inclusion and exclusion criteria are listed in figure 2. NFMA diagnosis is based on evidence for a pituitary macroadenoma on dedicated pituitary MRI and absence of clinical and biochemical signs of hormonal hypersecretion. Extension of the NFMA above the sellar diaphragm is a supplemental inclusion criterion since upward growth towards the optic nerves and chiasm is clinically relevant. In case of optic chiasm compression, current visual field defects must be excluded as this presents an indication for resection. Previous therapy with radiation, SSA or dopamine receptor agonists are exclusion criteria as these can

**Inclusion criteria**
- Clinically nonfunctioning pituitary macroadenoma extending above the sellar diaphragm
- Age ≥ 18 years
- Written informed consent

**Exclusion criteria**
- Optic chiasm compression with visual field defects
- Previous radiation therapy in the pituitary region
- Previous use of somatostatin analogues
- Use of dopamine receptor agonists in the past 6 months
- Hypersensitivity to somatostatin or similar peptides
- Obstructive neuroendocrine gut tumour
- Symptomatic cholelithiasis
- Pregnancy, lactation or wish to conceive
- Female patients of childbearing potential not using adequate contraception
- Any contraindication to perform MRI with gadolinium-based contrast agent (including implanted stimulation devices and severe claustrophobia)

**Figure 2** Eligibility criteria.

influence the primary outcome. Other exclusion criteria are in accordance with the summary of product characteristics of lanreotide (Somatuline AutoSolution).

## Study outline

Eligible patients are approached by their treating endocrinologist and permission is asked to be contacted by a trial staff member (TMB). The study is explained in full and written information is provided (online supplementary material S2 and S3). After obtaining written informed consent by a trial staff clinician, patient and NFMA characteristics are recorded (age, gender, medication use, laboratory and MRI results, current pituitary hormone deficiencies, prior NFMA treatment and further relevant medical history). At the screening visit, patients undergo a $^{68}$Ga-DOTATATE PET/CT of the head. Only patients with a PET-positive NFMA are randomised for treatment. Before start of treatment, a baseline visit is planned consisting of a fasting blood sample, short physical examination, quality of life questionnaire, structured interview and pregnancy test, if applicable. Pituitary MRI is only repeated at the baseline visit if time between the most recent MRI and planned start of study treatment exceeds three months or if the results provide any reason to repeat it. Study treatment consists of 18 subcutaneous injections administered every 28 days. Follow-up visits are planned at week 24 and week 48 (after the 6th and 12th injection, respectively), and the end-of-study visit takes place at week 72, approximately four weeks after the last injection. Study duration from first injection to last visit is thus 72 weeks. In case of withdrawal, a premature end-of-study visit is planned in the month after the last received injection. Study visits take place at the centre of inclusion (AMC, VUMC or LUMC) and will coincide with regular care assessments where possible. See figure 3 for the complete study schedule.

## $^{68}$Ga-DOTATATE PET/CT

$^{68}$Ga-DOTATATE PET/CT of the head is performed at either the AMC or the Netherlands Cancer Institute (Gemini ToF, Philips Medical Systems and Biograph mCT, Siemens Healthineers). $^{68}$Ga-DOTATATE preparation and quality control is performed according to a standard protocol as described elsewhere.[26] Images are acquired approximately 45 min after intravenous bolus injection of 100 MBq $^{68}$Ga-DOTATATE with 2.5–3 min per bed position. A low-dose CT scan is acquired for attenuation correction and anatomical correlation. The total estimated radiation exposure of the combined PET/CT is 3.1 mSv.[27] PET/CT and pituitary MRI are coregistered manually using Hybrid Viewer (Hermes Medical Solutions). On the fused PET/MR images, $^{68}$Ga-DOTATATE uptake is assessed by placement of a circular region of interest within the adenoma while maintaining a clear margin from normal pituitary tissue, and determining the mean standard uptake value ($SUV_{mean}$).[23] An $SUV_{mean}$ of >2 is considered as positive uptake. In the absence of literature on $^{68}$Ga-DOTATATE uptake in pituitary adenomas, this value has been chosen to reflect a level of uptake at least similar to that of the normal pituitary, based on a $^{68}$Ga-DOTATATE biodistribution study in 42 subjects demonstrating physiological pituitary uptake with a minimum $SUV_{mean}$ of 2.1.[28]

## Intervention

Forty-four patients with a $^{68}$Ga-DOTATATE PET-positive NFMA are randomised between treatment with either the extended-release formulation of lanreotide 120 mg (Somatuline AutoSolution, Ipsen Farmaceutica BV; known outside the Netherlands as Somatuline Autogel) without dose titration or placebo (consisting of 0.4 mL saline 0.9%), both administered every 28 days as a deep subcutaneous injection in the superior, external quadrant of the buttock. Study treatment consists of a total of 18 injections. The injections are administered by experienced and trained nurses, either at the AMC endocrine unit or at home via a homecare service (Eurocept Homecare). Administration details are recorded at each nurse visit. Lanreotide 120 mg is the highest dose currently

|  | STUDY PERIOD | | | | | | |
|---|---|---|---|---|---|---|---|
|  | Screen | Baseline | Treatment^a | | | | Close-out^b |
| Weeks |  | -1 / 0 | 0 | 24 | 48 | 68 | 72 |
| Visits | V₁ | V₂ |  | V₃ | V₄ |  | V₅ |
| **ENROLMENT:** |  |  |  |  |  |  |  |
| Eligibility screen | X |  |  |  |  |  |  |
| Informed consent | X |  |  |  |  |  |  |
| Baseline characteristics^c | X |  |  |  |  |  |  |
| Pregnancy test, if applicable^d | X | X |  |  |  |  |  |
| ⁶⁸Ga-DOTATATE PET/CT | X |  |  |  |  |  |  |
| Allocation |  | X |  |  |  |  |  |
| **INTERVENTIONS:** |  |  |  |  |  |  |  |
| Lanreotide |  |  | ◄———————► | | | |  |
| Placebo |  |  | ◄———————► | | | |  |
| **ASSESSMENTS:** |  |  |  |  |  |  |  |
| Pituitary MRI |  | X^e |  | X |  |  | X |
| Laboratory assessment^f |  | X |  | X | X |  | X |
| Physical examination^g |  | X |  | X | X |  | X |
| SF-36 survey |  | X |  | X | X |  | X |
| Concomitant medication |  | X |  | X | X |  | X |
| Adverse events |  | X | ◄—————————► | | | | X^h |

**Figure 3** Standard Protocol Items: Recommendations for Interventional Trials figure. Schedule of enrolment, interventions and assessments in the GALANT study. SF-36, 36-item Short Form Health Survey; PET, positron emission tomography. ^aTreatment consists of 18 injections administered every 28 days. ^bIn case of study withdrawal, a premature end-of-study visit is planned in the month after the last received injection. ^cPatient age, gender, medication use, most recent laboratory and pituitary MRI results, current pituitary hormone deficiencies, prior treatment and further relevant medical history. ^dOnly in female patients of childbearing potential. ^ePituitary MRI is only repeated if time between the most recent MRI and planned start of study treatment exceeds 3 months or if results give cause to repeat it. ^fEndocrine evaluation, fasting glucose, glycated haemoglobin (HbA1c), sodium, potassium, kidney function, liver enzymes and bilirubin; additional sample at baseline and close-out for alpha subunit assessment. ^gMeasurement of height, weight, blood pressure and pulse rate; more extensive examination if indicated. ^hUnresolved adverse events at time of end-of-study visit are followed-up until resolved or stabilised.

available. Treatment with this dose has shown to be safe and well-tolerated in two efficacy studies in enteropancreatic neuroendocrine tumours and acromegaly.[29 30] A safety review on SSA dose optimisation and a phase 2 randomised controlled trial specifically investigating lanreotide in different dosages did not demonstrate significant changes in adverse effects and tolerability with increasing dose. Higher doses were, however, related to greater efficacy.[31 32]

## Randomisation

Randomisation to treatment is done by means of a computer-generated randomisation list through Sealed Envelope (https://www.sealedenvelope.com/simple-randomiser/v1/lists) using blocked randomisation with an allocation ratio of 1:1 and block size of four.

## Allocation concealment and blinding

Randomisation is performed by the AMC trial pharmacy. The randomisation list is stored in the pharmacy trial file and will not be disclosed until the last participant has completed the trial. In case of emergency, the randomisation list can be accessed by the hospital pharmacist on call and the randomisation code can be broken if the safety of the patient requires knowledge of the assigned treatment (eg, in case of a suspected unexpected serious adverse reaction, SUSAR). Breaking of the code will be requested by an independent trial staff member to maintain allocation concealment to investigators assessing the outcomes. The study is blinded for participants, investigators and treating physicians. Because the placebo injections differ in appearance from the lanreotide (Somatuline AutoSolution) prefilled syringes, pharmacy employees preparing the treatment and nurses administering the injections cannot be blinded.[29] As the injections are administered into the superior, external quadrant of the buttock, patients will not see the syringes. Furthermore, patients have no earlier experience with lanreotide treatment to have expectations concerning the administration. To maintain blinding during transport and storage, the study medication is placed in an opaque bag within a sealed cardboard box with a 'blinded study medication' warning message.

## Assessments during the study
### Pituitary MRI

Pituitary MRIs are performed regularly as part of standard care of NFMA patients. Study participation is timed in such a way that visits coincide with regular MRI planning where possible. MRI scans are performed on a 1.5 or 3 Tesla scanner, following a pituitary-specific protocol that includes T1-weighted coronal and sagittal acquisitions before and after gadolinium administration, a T2-weighted coronal sequence and preferably a volumetric 3DT1-weighted sequence.

### Laboratory tests

Regular laboratory assessments are also part of standard care. During the study, laboratory tests will include an endocrine evaluation (TSH, free T4, cortisol, IGF-1, prolactin and testosterone or oestradiol), fasting glucose, glycated haemoglobin (HbA1c), sodium, potassium, kidney function, liver enzymes and bilirubin. At baseline and week 72, additional blood samples are obtained for alpha subunit assessment. Leftover serum and plasma will

be stored according to local regulatory guidelines until publication of results.

### Physical examination

Measurement of height, weight, blood pressure and pulse rate takes place at each visit after randomisation. More extensive examination is only performed if indicated.

### Quality-of-life questionnaire

The 36-item Short Form Health Survey (SF-36), validated Dutch translation, is used to assess quality of life at study visits.[33] The SF-36 comprises 36 items that generate eight component scores: physical functioning, role limitations due to physical health problems, bodily pain, general health perceptions, vitality, social functioning, role limitations due to emotional problems and general mental health. Scores are expressed on a 0–100 scale with higher scores indicating better quality of life for that component.

### Structured interview

The structured interview at visits during treatment consists of health-related questions based on possible side-effects of lanreotide, other signs and symptoms experienced between visits, and changes in medication use.

### Adverse events

Adverse events (AEs) can be reported throughout the study period and will be inquired about specifically at study and nurse visits. An AE is defined as any undesirable finding or experience occurring to a subject during the study, whether or not considered related to study treatment. The type of AE, times of onset and resolution, severity and possible relationship with lanreotide will be recorded.

### Concomitant care and post-trial follow-up

Participants are not allowed to use medication or participate in other trials aimed at adenoma reduction. Otherwise, patient care is continued as usual. Participants may withdraw from the study for any reason at any time. In case of development of visual field impairments or signs pointing to cranial nerve injury, patients will undergo additional pituitary MRI and visual field examination as is standard practice. If intervention is deemed necessary, study treatment is stopped. Patients can also be withdrawn based on the investigator's or treating specialist's judgement, for example, in case of an AE that could jeopardise patient safety. Premature study withdrawal will have no effect on regular patient care.

Lanreotide can inhibit the secretion of insulin and glucagon. Patients on antidiabetic therapy are therefore advised to monitor their blood glucose levels more closely after start of study treatment and have their medication adjusted accordingly. Concomitant use of bradycardia inducing drugs (eg, beta blockers) may have an additive effect on the slight reduction of heart rate associated with lanreotide and dose adjustments may be necessary. Intestinal absorption of most notably ciclosporin can be reduced, requiring close review of therapeutic levels.

Participants are instructed to notify the research team if they have been prescribed a new drug by their general practitioner or specialist during the study.

After completion of study treatment or premature withdrawal, standard care is continued. Patients who leave the study for medical reasons/AEs will be followed until the interfering condition has resolved or reached a stable state. All participants will be followed until completion of the trial and will be notified of the results.

### Primary outcome

The primary outcome is the absolute change in cranio-caudal tumour diameter in millimetres from baseline to week 72 (or last available postbaseline measurement). Tumour size will be determined on pituitary MRI by two neuroradiologists blinded to treatment allocation and scan chronology. The measurements of both radiologists are averaged for the analysis. In case of a between-reader difference of ≥10% or ≥2 mm, the images will be offered for reassessment. In case of a persistent difference of ≥10% or ≥2 mm, a third neuroradiologist will review the images and the measurements of the three radiologists are averaged for the analysis. A change in tumour size of ≥2 mm is considered clinically relevant, as such a difference is sufficient to prevent complications related to tumour growth. Even a small increase in cranio-caudal diameter can be reason for intervention via surgery or radiotherapy to prevent or relieve compression of the optic nerves or chiasm.[34] Moreover, the value of 2 mm is commonly used as cut-off to reliably detect a change in tumour size on pituitary MRI.[35–37]

### Secondary outcomes
#### Change in tumour volume

As with the primary outcome, the change in tumour volume between baseline and week 72 (or last available postbaseline measurement) is determined by two neuroradiologists or trained and supervised study personnel using pituitary MRI. Tumour volume in cubic millimetres will be calculated using a semiautomatic segmentation tool in the coronal plane. On each slice, the tumour is delineated and the surface area multiplied by the slice thickness. The total tumour volume is determined by summing up the slice volumes. The slice-by-slice segmentation method is preferred to the traditional geometric formula (ie, ½ × length × width × height) due to the frequent irregular shape and invasion of NFMA, especially if it concerns a postoperative remnant.[38] In case of a between-reader difference of ≥10%, the images will be offered for reassessment. In case of a persistent difference of ≥10%, a third neuroradiologist or experienced study staff member will review the images. A change in tumour volume of ≥20% will be considered clinically significant.[30]

#### Time to progression

The presence or absence of tumour progression will be based on clinically significant change in tumour volume between baseline and subsequent pituitary MRIs. Tumour

progression is defined as an increase in volume of ≥20%, absence of progression comprises both tumour shrinkage (volume decrease ≥20%) and stable disease (volume change <20%).[30]

### Change in quality of life

The change in quality of life is based on the SF-36 questionnaire component scores obtained at the study visits.

### Number of (serious) AEs

As a safety endpoint, the number and type of AEs and serious AEs (SAEs) recorded during the study will be compared between the groups.

### Sample size

The study is powered on a between-group difference in primary outcome of ≥2 mm. In order to detect a mean difference of 2 mm with a SD of 1.9 mm, we need to randomise 16 patients per group based on a two-sided independent t-test with 80% power and 5% type I error risk (nQuery Advisor 7.0, Statsol, Boston, Massachusetts, USA). The final analysis of the primary endpoint will be performed by analysis of covariance (ANCOVA). While ANCOVA may reduce the required sample size, a reliable estimation of rho is not possible for this study and a more conservative t-test-based power analysis is therefore less likely to result in an underpowered study.[39] Taking into account an observed dropout rate of around 25%, as a result of withdrawals due to AEs or need for intervention, the sample size has been increased to 22 patients per group (substantial amendment, protocol version 5.0). Dropout is defined as failure to complete the 72 weeks of treatment due to any reason. Only patients with a $^{68}$Ga-DOTATATE PET-positive NFMA are randomised for treatment. Based on previous reports that about two-thirds of NFA patients showed increased adenoma uptake using $^{111}$In-DTPA-octreotide scintigraphy[10 14 15 40] and considering the superior sensitivity of $^{68}$Ga-DOTATATE PET/CT, we expected to enrol a maximum of 66 patients in order to randomise 44 patients.

### Statistical analyses

Data of continuous variables will be summarised as either means with SD or medians with IQR, according to distribution. Normality of distribution will be assessed visually with histograms, Q-Q plots and using the Shapiro-Wilk test. Data of categorical variables will be reported as incidence rates (number and percentage). Baseline characteristics and $^{68}$Ga-DOTATATE PET/CT results of all included subjects will be reported. Data on recruitment and dropout will also be reported. Differences between patients with positive and negative $^{68}$Ga-DOTATATE uptake will be evaluated. Group differences in continuous variables are analysed using Student's t-test or Mann-Whitney U test, depending on data distribution. Group differences in categorical variables are analysed using either the $\chi^2$ or Fisher's exact test, as appropriate. In principle, since patients are randomly allocated to treatment, differences in baseline characteristics between the groups

should not be analysed. When performing linear regression analyses or linear mixed effects models, assumptions for linearity, distribution and/or homoscedasticity will be checked for each model using scatter plots and residual analysis.

Outcome data will be analysed according to intention-to-treat (ITT). The ITT population is defined as all randomised patients having received at least one study injection. In case of withdrawal after treatment initiation, effort is made to plan a premature end-of-study visit to obtain outcome data for the ITT analyses, unless assessments were performed within the previous 4 weeks. In case of incomplete/missing outcome data, the assumption that data are missing at random (MAR) will be checked. If the MAR assumption is plausible, a main analysis using all observed data is performed (eg, a mixed model). This main analysis does not require the inclusion of subjects that lack postbaseline assessments.[41] Sensitivity analyses including all ITT subjects are then performed to explore the impact of departures from this assumption.[41] This includes an analysis using multiple imputation in case of missing outcome data.[42]

The difference between groups in primary outcome and secondary outcome change in tumour volume will be analysed using one-way ANCOVA, controlling for baseline adenoma size. For these two endpoints, a separate per-protocol efficacy analysis will be performed, including only those subjects who have completed all 18 study injections and final MRI. Additionally, the proportion of patients with a clinically significant change in tumour size (ie, ≥2 mm change in cranio-caudal size or ≥20% change in tumour volume) is reported. Time to tumour progression is analysed using the Kaplan-Meier method. Missing cases due to withdrawal without outcome data will be censored. Differences in distribution and percentage of censoring between groups will be evaluated. Between-group differences in time to progression are analysed using the stratified logrank test, with stratification for presence or absence of tumour growth at baseline. To quantify the difference, the HR and CIs are estimated using the Cox proportional-hazards model. The SF-36 component scores will be presented as spider charts. The mean imputation method will be used to replace missing values. Results will be compared with Dutch population reference values[33] and the change in relevant component scores are compared between the groups. All patients who received at least one study injection will be included in the safety endpoint analysis of AEs independent of postbaseline assessments.

As the study remains blinded until completion, no interim analyses are planned. There are no prespecified subgroups. A p-value <0.05 will be considered statistically significant. In case of multiple testing, Bonferroni correction will be applied where appropriate. Besides the reporting of p-values, 95% CIs around point estimates will be provided for more useful interpretation of results. Analyses will be done using SPSS (IBM, V.25 or later if available) and/or RStudio (R Foundation for Statistical

Computing, V.3.6.1, 2019-07-05, or later if available). Before outcome analyses are initiated, the statistical analysis plan will be finalised.

## Data management

Data are collected using Castor Electronic Data Capture.[43] The electronic case report forms (CRFs) have built-in data validity and value range checks. Source documents include medical records, records of the Eurocept homecare service, SF-36 questionnaires and paper CRFs documenting injections administered at the AMC. All subject data are pseudonymised with a study code. The subject identification log which links subjects to the code is kept in a trial file only accessible to study personnel. All research data will be stored for 15 years. Leftover serum and plasma from blood samples obtained for standard care during the study will be stored according to local regulatory guidelines until publication of the results and then destroyed. Data will be handled in accordance with the principles of the General Data Protection Regulation. All data management procedures are detailed in a separate data management plan.

## Safety

The most commonly expected side-effects of lanreotide are gastrointestinal disorders (eg, diarrhoea, abdominal pain and nausea, usually mild/moderate and transient), cholelithiasis (often asymptomatic) and injection site reactions. Additionally, lanreotide may induce (mild) hypoglycaemia or hyperglycaemia due to inhibition of insulin and/or glucagon secretion. Because of surveillance at regular intervals during the study, the risks are small. Throughout the study, AEs reported by the patient or observed by the investigators or healthcare providers are recorded in the medical file and study database. All SAEs including SUSARs are reported to the Medical Research Ethics Committee (MREC) of the AMC. In addition, an annual safety report is submitted.

## Monitoring

The trial is monitored by the Clinical Research Unit of the AMC. Participating centres agree to allow monitoring by providing access to source data and documents as required. Every participating centre is subjected to a start-up visit after three included patients, a second site-visit after approximately 10 inclusions and a combined monitoring and site close-out visit after completion of data collection. The monitor confirms all written informed consents, checks all SAEs and investigates data collection and data quality for a random subset of patients. The frequency and intensity of monitoring can be changed based on findings during visits. Monitoring procedures are detailed in a separate monitoring plan.

## Patient and public involvement

Patients were not involved in the design of this study. The study was highlighted in 'Hyponieuws', the magazine of the Dutch Pituitary Foundation, which is the Dutch pituitary patient platform.

## ETHICS AND DISSEMINATION

The study protocol has been approved by the accredited MREC of the AMC and by the Dutch competent authority. Permission for execution of the study is obtained at each participating site. Substantial amendments to the protocol are submitted for approval and communicated to participating centres and trial registries. The study is conducted in accordance with the Declaration of Helsinki, Good Clinical Practice guidelines and local regulatory requirements. Participant insurance is provided by the AMC for all participating subjects.

Regardless of the outcome, study results will be submitted for publication in a peer-reviewed international medical journal and presented at (inter)national conferences. The International Committee of Medical Journal Editors guidelines for authorship will be followed. The AMC, as sponsor, will remain owner of all data and have all publication rights. Full protocol, data sets and statistical analysis plan will be available from the corresponding author on reasonable request after publication of the final results.

## Trial status

Recruitment started on 3 November 2015. At the time of preparing this manuscript, inclusion has been completed. The last visit of the last patient is expected in the first half of 2021. The current protocol version is V5.0 (30 November 2018).

## DISCUSSION

The GALANT study is the first double-blind and placebo-controlled intervention trial in NFMA patients, investigating the effect of lanreotide 120 mg on NFMA size. Treatment duration is 72 weeks, in line with the slow average NFA growth rate of around 1 mm/year.[4 35] An additional strength of the study design is the use of $^{68}$Ga-DO-TATATE PET/CT to select only those patients with an SSTR-positive adenoma for treatment. Although in a clinical setting dose titration might mitigate side-effects and prevent treatment cessation, use of the highest dose currently available is justified in order to prove efficacy for a possible new therapeutic indication and ensures data homogeneity. Furthermore, treatment with this dose has shown to be safe and well tolerated in earlier trials.[29–31] The study is powered to detect a between-group difference of ≥2 mm change in cranio-caudal tumour size after treatment, a clinically relevant outcome. Consistent and unbiased tumour measurements are ensured through assessment by at least two neuroradiologists blinded for treatment allocation and scan chronology. If lanreotide proves effective in reducing or controlling tumour size, this would present an important new treatment option for NFMA patients.

**Author affiliations**
[1]Department of Endocrinology and Metabolism, Amsterdam Gastroenterology Endocrinology Metabolism, Amsterdam UMC, location AMC, University of Amsterdam, Amsterdam, The Netherlands

[2]Department of Internal Medicine, Section of Endocrinology, Amsterdam UMC, location VUMC, VU University, Amsterdam, The Netherlands

[3]Department of Radiology and Nuclear Medicine, Amsterdam UMC, location AMC, University of Amsterdam, Amsterdam, The Netherlands

[4]Department of Nuclear Medicine, Netherlands Cancer Institute, Amsterdam, The Netherlands

[5]Department of Neurosurgery, Neurosurgical Centre Amsterdam, Amsterdam UMC, location AMC, University of Amsterdam, Amsterdam, The Netherlands

[6]Department of Medicine, Division of Endocrinology, and Centre for Endocrine Tumors Leiden (CETL), Leiden University Medical Centre, Leiden, The Netherlands

[7]Department of Internal Medicine, Northwest Clinics, Alkmaar, The Netherlands

[8]Department of Internal Medicine, Medical Spectrum Twente, Enschede, The Netherlands

[9]Department of Clinical Epidemiology, Biostatistics and Bioinformatics, Amsterdam UMC, location AMC, University of Amsterdam, Amsterdam, The Netherlands

**Contributors** TMB, MLD, JB, CBLM, MPMS, MWTT, PHB and EF participated in the design of the study. EF, MLD, MPMS and AP are principal investigators at the participating study centres. EF supervises the study in name of the sponsor (AMC). MLD, JH, AP, NRB, SS, RGV, PHB and EF assist with the referral and inclusion of patients. TMB drafted the manuscript with help of PHB and EF. All authors reviewed and approved the final manuscript.

**Funding** This work is supported by Ipsen Farmaceutica BV (the Netherlands). The GALANT study is investigator-initiated; the Academic Medical Centre (AMC) is the sponsor of the study. The funding party has played no role in the final study design or study conduct nor in the preparation of this manuscript or the decision to submit it for publication. The AMC, as sponsor, will remain owner of all data and have all publication rights.

**Competing interests** MLD received a honorarium for chairing a symposium sponsored by Ipsen Farmaceutica BV in 2019.

**Patient and public involvement** Patients and/or the public were not involved in the design, or conduct, or reporting or dissemination plans of this research.

**Patient consent for publication** Not required.

**Ethics approval** Medical Ethics Committee Academic Medical Centre, of the Amsterdam University Medical Centres (reference: NL 52821.018.15).

**Provenance and peer review** Not commissioned; externally peer reviewed.

**ORCID iD**
Tessel M Boertien http://orcid.org/0000-0002-1285-4834

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
