## [Reviewer comments · BMJ Open]

ARTICLE DETAILS

TITLE (PROVISIONAL)	The GALANT trial: study protocol of a randomised placebo-controlled trial in patients with a 68Ga-DOTATATE PET-positive, clinically nonfunctioning pituitary macroadenoma on the effect of lanreotide on tumour size
AUTHORS	Boertien, Tessel; Drent, Madeleine; Booij, Jan; Majoie, Charles; Stokkel, Marcel; Hoogmoed, Jantien; Pereira, Alberto; Biermasz, Nienke; Simsek, Suat; Groote Veldman, Ronald; Tanck, Michael; Fliers, Eric; Bisschop, Peter

VERSION 1 – REVIEW

REVIEWER	Mauro Cives University of Bari, Italy Speaker fees from Ipsen
REVIEW RETURNED	22-Mar-2020

GENERAL COMMENTS	This is a well-written study protocol investigating the anti-tumor activity of lanreotide in patients with nonfunctioning pituitary macroadenoma. The research question is clearly detailed and scientifically sound. The methods are appropriate. I have only a few comments: 1) Please clarify lines 177-178. 2) There is a typo at line 281. Please, edit.
---

REVIEWER	Dimitrios Priftakis Institute of Nuclear Medicine, University College London Hospital, United Kingdom
REVIEW RETURNED	01-Apr-2020

GENERAL COMMENTS	This protocol describes the study design of a randomised placebo-controlled trial in patients with 68Ga-DOTATATE PET-positive nonfunctioning pituitary adenoma on the effect of lanreotide on tumour size (GALANT trial). Although the utility of long-acting release somatostatin analogs is established in patients with functioning adenomas (particularly patients with acromegaly), there is not much information in the literature about outcomes in non-functioning pituitary adenomas. A previous similar study by Fusco et al. in 2012, referenced by the authors, showed favorable response of patients with nonfunctioning pituitary adenoma treated with octreotide LAR, more so in those with positive baseline somatostatin receptor scintigraphy, a method less sensitive than 68Ga-DOTATATE PET used in the current study. Therefore, the purpose of the study is original and with interesting implications for the treatment of these patients. The study design is robust and the methodology is thorough and
---

	described in detail. There is a satisfactory mechanism for the placebo administration and the blinding of the results. The authors have set measurable and achievable primary and secondary outcomes. They have provided acceptable safety precautions and provision of patient withdrawal to avoid harm in accordance with good clinical practice. Minor observations on study design choices that, in my opinion, could potentially limit the results of the study, and should be clarified by the authors, are the following:  1. The inclusion of both surgery-naive patients and patients with a postoperative remnant. This could lead to heterogeneous results, and although the target of enrolled patients is within acceptable range, there might not be enough for a subgroup analysis. Moreover, there is no criterion for size of the remnant (a small remnant might grow slower than a large adenoma?). 2. The use of two different PET/CT scanners might insert variability in the SUV measurements, if there has been no provision for cross-calibration or at least proof of comparability. 3. The threshold of SUV_{mean} > 2 as criterion for positive 68Ga-DOTATATE uptake has a reasonable rationale but is weakly supported by data (expected as there is not much information in the literature). In your team's recently published study (Boertien, T.M., Booij, J., Majoie, C.B.L.M. et al. 68Ga-DOTATATE PET imaging in clinically non-functioning pituitary macroadenomas. European J Hybrid Imaging 4, 4 (2020). https://doi.org/10.1186/s41824-020-0073-3), you showed a high positivity rate 91.9% using this threshold, which you attributed to the high sensitivity of the method, but could actually mean that the chosen threshold is too low. It would be useful to explore if the outcome of lanreotide treatment is better in patients with higher baseline uptake. If this is not true, the use of 68Ga-DOTATATE PET as a baseline imaging method to select patients for lanreotide treatment might not be justified. 4. The definition of tumour progression is an increase in volume of >20% (line 310-312). Can you please add the reference for this? 5. Line 449, according to the reference you give (Dekkers et al. 2007) the average growth rate is 0.6 mm/year. Of course, the study will provide specific data for the studied population and a retrospective evaluation of the adequacy of the study duration. I am looking forward to your feedback and to the results of your interesting, clinically relevant research.
--	--

VERSION 1 – AUTHOR RESPONSE

Reviewer 1, dr. Mauro Cives

This is a well-written study protocol investigating the anti-tumor activity of lanreotide in patients with nonfunctioning pituitary macroadenoma. The research question is clearly detailed and scientifically sound. The methods are appropriate.

We thank dr. Cives for this positive assessment of our manuscript.

I have only a few comments:

1. Please clarify lines 177-178.

Thank you for this suggestion. We have rephrased the text to better clarify the choice for an SUV_{mean} cut-off of >2 (lines 177-181 in the revised, marked manuscript):

An SUV_{mean} of >2 is considered as positive uptake. In the absence of literature on ^{68}Ga -DOTATATE uptake in pituitary adenomas, this value has been chosen to reflect a level of uptake at least similar to that of the normal pituitary, based on a ^{68}Ga -DOTATATE biodistribution study in 42 subjects demonstrating physiological pituitary uptake with a minimum SUV_{mean} of 2.1 (Shastri et al. 2010).

2. There is a typo at line 281. Please, edit.

We appreciate the attentive reading of our manuscript. We have reviewed line 281 (line 284 in the revised, marked manuscript) and could, however, not distinguish a typo:

The primary outcome is the absolute change in cranio-caudal tumour diameter in millimetres from

The text therefore remains unchanged. Please note that we have used spelling according to UK English.

Reviewer 2, dr. Dimitrios Priftakis

This protocol describes the study design of a randomised placebo-controlled trial in patients with ^{68}Ga -DOTATATE PET-positive nonfunctioning pituitary adenoma on the effect of lanreotide on tumour size (GALANT trial). Although the utility of long-acting release somatostatin analogs is established in patients with functioning adenomas (particularly patients with acromegaly), there is not much information in the literature about outcomes in non-functioning pituitary adenomas. A previous similar study by Fusco et al. in 2012, referenced by the authors, showed favorable response of patients with nonfunctioning pituitary adenoma treated with octreotide LAR, more so in those with positive baseline somatostatin receptor scintigraphy, a method less sensitive than ^{68}Ga -DOTATATE PET used in the current study. Therefore, the purpose of the study is original and with interesting implications for the treatment of these patients.

The study design is robust and the methodology is thorough and described in detail. There is a satisfactory mechanism for the placebo administration and the blinding of the results. The authors have set measurable and achievable primary and secondary outcomes. They have provided acceptable safety precautions and provision of patient withdrawal to avoid harm in accordance with good clinical practice.

We thank dr. Priftakis for marking our study as original and for the overall positive appraisal of the study design.

Minor observations on study design choices that, in my opinion, could potentially limit the results of the study, and should be clarified by the authors, are the following:

- 1. The inclusion of both surgery-naïve patients and patients with a postoperative remnant. This could lead to heterogeneous results, and although the target of enrolled patients is within acceptable range, there might not be enough for a subgroup analysis. Moreover, there is no criterion for size of the remnant (a small remnant might grow slower than a large adenoma?).**

Thank you for this observation. The decision to include both surgery-naïve patients and patients with a postoperative remnant follows the notion that both would benefit from a medical treatment option. The size criterion, as described in line 135 and figure 2, is the same for all patients: a macroadenoma (i.e. diameter $\geq 1\text{cm}$) extending above the sellar diaphragm. Propensity for growth and growth rate of nonfunctioning macroadenoma (NFMA) have been shown to be unrelated to tumour size (Honegger et al. 2008; Huang and Molitch 2018). Furthermore, the proportion of patients with tumour progression over time is similar for NFMA followed-up conservatively and postoperative remnants not treated with adjuvant radiotherapy (Chen et al. 2012). We therefore feel that inclusion

of patients both surgery-naïve and with a postoperative remnant will not lead to heterogeneous results.

2. The use of two different PET/CT scanners might insert variability in the SUV measurements, if there has been no provision for cross-calibration or at least proof of comparability.

Thank you for identifying this potential limitation. We agree that SUV measurements are scanner dependent to a certain extent (Adams et al. 2010). Only the first four included patients have undergone the PET/CT scan at the Netherlands Cancer Institute, since at that time, ⁶⁸Ga-DOTATATE PET scanning was not available yet in the AMC. Both centres employed the same PET scanner (Philips Gemini ToF). However, in the course of the study, the Philips Gemini ToF in the AMC has been replaced with the Siemens Biograph mCT scanner. Unfortunately, no phantom study has been performed in this process. Apart from a change in scanner, the protocol concerning patient preparation, image acquisition and reconstruction has remained the same. Furthermore, all scans have been assessed using the same software (Hybrid Viewer (Hermes Medical Solutions)) and by the same authors (TB and JB).

As at the time of submission of the manuscript patient inclusion has been completed, we can elaborate somewhat on the PET results. Of the 49 patients included, 17 underwent PET/CT on the Philips Gemini ToF scanner (SUV_{mean} 6.6 ±3.2, mean ±SD [range 2.61-13.56]) and 32 on the Siemens Biograph mCT scanner (SUV_{mean} 4.8 ±3.2, mean ±SD [range 0.40-14.37]). This difference in SUV_{mean} did not reach statistical significance ($p = 0.069$, Students t -test) and can be fully explained by the unequal ratio of negative/positive uptake cases between the scanners, as incidentally all negative cases were scanned on the Biograph mCT.

3. The threshold of SUV_{mean} > 2 as criterion for positive ⁶⁸Ga-DOTATATE uptake has a reasonable rationale but is weakly supported by data (expected as there is not much information in the literature). In your team's recently published study (Boertien, T.M., Booi, J., Majoie, C.B.L.M. et al. ⁶⁸Ga-DOTATATE PET imaging in clinically non-functioning pituitary macroadenomas. *European J Hybrid Imaging* 4, 4 (2020). [We thank dr. Priftakis for raising this important issue and for referring to our recently published study. We acknowledge that the SUV_{mean} threshold of >2 is based on limited data \(see also our response to comment 1 of Reviewer 1 and the revised text in lines 177-181\). The high rate of positive uptake observed during the study was quite unexpected. Although uptake above the used threshold does indicate presence of somatostatin receptors, it is uncertain at this point if this level is sufficient for a favourable treatment effect of somatostatin analogues. It will therefore indeed be valuable to explore whether treatment effect is related to the level of ⁶⁸Ga-DOTATATE uptake when the trial is completed.](https://eur04.safelinks.protection.outlook.com/?url=https%3A%2F%2Fdoi.org%2F10.1186%2Fs41824-020-0073-3&data=02%7C01%7Ct.m.boertien%40amsterdamumc.nl%7C622d47f7cfc2444f19e408d8113a5596%7C68dfab1a11bb4cc6beb528d756984fb6%7C0%7C0%7C637278289526139507&sdata=Jm4gJ1HIVwhN4gCIDjMUtkuOakq%2FMWPhw%2FBXAJU%2F898%3D&r eserved=0), you showed a high positivity rate 91.9% using this threshold, which you attributed to the high sensitivity of the method, but could actually mean that the chosen threshold is too low. It would be useful to explore if the outcome of lanreotide treatment is better in patients with higher baseline uptake. If this is not true, the use of ⁶⁸Ga-DOTATATE PET as a baseline imaging method to select patients for lanreotide treatment might not be justified.20% (line 310-312). Can you please add the reference for this?

The definition of tumour progression is based on a clinically significant change in tumour volume of $\geq 20\%$, a generally accepted cut-off (Caron et al. 2014), described in the previous paragraph of the manuscript (line 309 of the revised, marked manuscript). We have followed the recommendation to add this reference in the paragraph concerning tumour progression as well (line 315 in the revised, marked manuscript). We have also added the words “*clinically significant*” in line 312 to clarify the definition.

5. Line 449, according to the reference you give (Dekkers et al. 2007) the average growth rate is 0.6 mm/year. Of course, the study will provide specific data for the studied population and a retrospective evaluation of the adequacy of the study duration.

This is indeed an important point. The natural growth rate of NFMA is known to be quite variable (Dekkers et al. 2007; Huang and Molitch 2018), which makes determining appropriate study duration for this type of pituitary adenoma difficult. A study duration of at least 12 months is considered adequate to assess size outcome in acromegaly patients on somatostatin analogue treatment (Fleseriu 2011; Marazuela et al. 2018). We have decided upon a treatment duration of 72 weeks to strike a balance between proving efficacy and feasibility, and will certainly reflect upon this in the discussion of our results.

6. I am looking forward to your feedback and to the results of your interesting, clinically relevant research.

Thank you for the interest in our study and for the thorough review. We hope we have addressed all issues satisfactorily.

Besides changes to the manuscript discussed in the response to the reviewers above, we have corrected two typos discovered during the revision process (line numbers correspond to those in the marked copy of the manuscript):

- line 135: *surgery-naive* changed to *surgery-naïve*
- line 476: *participating* changed to *participating*
- line 629: *macroacdenoma* changed to *macroadenoma*

References

- Adams, Michael C., Timothy G. Turkington, Joshua M. Wilson, and Terence Z. Wong. 2010. “A Systematic Review of the Factors Affecting Accuracy of SUV Measurements.” *American Journal of Roentgenology* 195(2): 310–20. <http://www.ajronline.org/doi/10.2214/AJR.10.4923>.
- Bettinardi, V., I. Castiglioni, E. De Bernardi, and M. C. Gilardi. 2014. “PET Quantification: Strategies for Partial Volume Correction.” *Clinical and Translational Imaging* 2(3): 199–218. <http://link.springer.com/10.1007/s40336-014-0066-y>.
- Caron, Philippe J. et al. 2014. “Tumor Shrinkage With Lanreotide Autogel 120 Mg as Primary Therapy in Acromegaly: Results of a Prospective Multicenter Clinical Trial.” *The Journal of Clinical Endocrinology & Metabolism* 99(4): 1282–90. <https://academic.oup.com/jcem/article/99/4/1282/2537335>.
- Chen, Yong et al. 2012. “Natural History of Postoperative Nonfunctioning Pituitary Adenomas: A Systematic Review and Meta-Analysis.” *Neuroendocrinology* 96(4): 333–42. <http://www.ncbi.nlm.nih.gov/pubmed/22687984>.
- Dekkers, Olaf M. et al. 2007. “The Natural Course of Non-Functioning Pituitary Macroadenomas.” *European Journal of Endocrinology* 156(2): 217–24.

- Fleseriu, Maria. 2011. "Clinical Efficacy and Safety Results for Dose Escalation of Somatostatin Receptor Ligands in Patients with Acromegaly: A Literature Review." *Pituitary* 14(2): 184–93. <http://link.springer.com/10.1007/s11102-010-0282-z>.
- Honegger, Juergen et al. 2008. "Growth Modelling of Non-Functioning Pituitary Adenomas in Patients Referred for Surgery." *European Journal of Endocrinology* 158(3): 287–94. <https://eje.bioscientifica.com/view/journals/eje/158/3/287.xml>.
- Huang, Wenyu, and Mark E. Molitch. 2018. "Management of Nonfunctioning Pituitary Adenomas (NFAs): Observation." *Pituitary* 21(2): 162–67. <http://dx.doi.org/10.1007/s11102-017-0856-0>.
- Marazuela, Monica, Ana M. Ramos-Leví, Patricia Borges de Souza, and Maria Chiara Zatelli. 2018. "Is Receptor Profiling Useful for Predicting Pituitary Therapy?" *European Journal of Endocrinology* 179(5): D15–25. <https://eje.bioscientifica.com/view/journals/eje/179/5/EJE-18-0549.xml>.
- Shastry, Manu et al. 2010. "Distribution Pattern of 68Ga-DOTATATE in Disease-Free Patients." *Nuclear Medicine Communications* 31(12): 1. <https://insights.ovid.com/crossref?an=00006231-900000000-99763>.